# Zero-Field Slow Magnetic Relaxation in Binuclear Dy Acetylacetonate Complex with Pyridine-N-Oxide

**Valeriya P. Shtefanets** [1,2], **Gennady V. Shilov** [1], **Denis V. Korchagin** [1], **Elena A. Yureva** [1], **Alexei I. Dmitriev** [1], **Mikhail V. Zhidkov** [1], **Roman B. Morgunov** [1], **Nataliya A. Sanina** [1,*] **and Sergey M. Aldoshin** [1]

[1] Federal Research Center of Problems of Chemical Physics, Medicinal Chemistry RAS, prosp. Acad. Semenova, 1, 142432 Chernogolovka, Russia

[2] Moscow Institute of Physics and Technology, National Research University, 9 Institutskiy per., 141701 Dolgoprudny, Russia

[*] Correspondence: sanina@icp.ac.ru

**Abstract:** A new complex $[Dy(C_5H_7O_2)_3(C_5H_5NO)]_2 \cdot 2CHCl_3$ (**1**) has been synthesized by the reaction of pyridine-N-oxide with dysprosium (III) acetylacetonate in an n-heptane/chloroform mixture (1/20). X-ray data show that each dysprosium atom is chelate-like coordinated by three acetylacetonate ligands and the oxygen atom from two bridging molecules of pyridine-N-oxide, which unite the dysprosium atoms into a binuclear complex. Static (constant current) and dynamic (alternating current) investigations and ab initio calculations of the magnetic properties of complex **1** were performed. The complex was shown to exhibit a frequency maximum under alternating current. At temperatures above 10 K, the maximum shifts to a higher frequency, which is characteristic of SMM behavior. It is established that the dependence of $\ln(\tau)$ on $1/T$ for the relaxation process is nonlinear, which indicates the presence of Raman relaxation mechanisms, along with the Orbach mechanism.

**Keywords:** single molecule magnets; complexes of rare earth metals; pyridine-N-oxide; X-ray diffraction analysis; IR-spectroscopy; magnetometry

## 1. Introduction

Rare-earth metals have been widely studied as candidates for applications in information quantum processing and high-density data storage on a molecular and atomic level [1–5]. The design of rare-earth single-molecule magnets (SMMs) that are able to act as magnets on a molecular level is particularly urgent and is of particular interest to researchers [6]. In mononuclear complexes of lanthanides with polyoxometallates, phthalocyanines, and β-diketones, and organometallic lanthanide SMMs, a strong single-ion anisotropy is the most important characteristic for exhibiting SMM behavior. However, QTM quick relaxation is the prevailing feature of lanthanide-based SMMs [7]. In particular, in mononuclear SMMs, the QTM contribution to total relaxation increases with decreasing temperature [8–12]. Therefore, the development of approaches to reduce QTM relaxation at low temperatures is still the key problem in rare-earth-based SMM design.

In this context, the design of rare-earth polynuclear clusters for the decrease of QTM (or for its suppression, like in the case of rare-earth Gd and Dy [13,14] binuclear complexes) at the expense of increasing the magnetic interaction between the lanthanide ions is being developed intensely. Some results have already been obtained, including the presence of ultrahigh blocking barriers [15–17] and interesting magnetic phenomena [18,19] in polynuclear rare-earth SMMs.

Among lanthanide-based SMMs, complexes containing Dy(III) with various ligands are of particular interest, i.e., both mononuclear complexes [20–26] that are characterized by the shortcomings we have discussed above and polynuclear complexes [18,27–33]. Some mononuclear Dy-based SMMs exhibit a high effective energetic barrier of magnetic

relaxation (1815 K [34] compared to the transition metal complexes, with the highest value of 325 K for the iron complex [35]).

For the synthesis of rare-earth SMMs, β-diketones (acetylacetonates (acac), hexafluorine acetylacetonates (hfac)), and their derivatives are generally used [36–39]. For example, mononuclear complexes of rare-earth metals with acac, with the metal coordinated by three acac ligands and two water molecules, exhibit magnetic relaxation at 8 K and have an energetic barrier up to 66.1 K, while diluting with a non-magnetic metal causes the suppression of quantum tunneling [40]. The binuclear complex, consisting of six deprotonated acac ligands, two coordinated water molecules, and two Dy(III) cations, exhibits a frequency dependence of magnetic susceptibility at temperatures below 6 K [41]. Rare-earth complexes with hfac do not always exhibit SMM properties, but even with these materials magnetic relaxation is observed below 4 K, therefore they are becoming popular as precursors for growing crystals with ligands of the combined type. For instance, binuclear Dy(III) complexes with PyNO demonstrate magnetic relaxation at 1.4 K [42]. Although over the past few years, pyridine-N-oxide derivatives have been used as ligands to prepare Dy(III)-containing SMMs, the number of reported examples is still limited [43]. Substitutions in the PyNO ring change the crystal properties inconsiderably, with magnetic relaxation at temperatures below 2 K [44]. These results are important for the development of technological applications with rare-earth SMMs as key molecular components forming coatings on the adapted supports. Earlier, Dy-based SMMs with pyridine-N-oxide were reported [42], which are able to be physically sorbed on the surface as thick films using sublimation methods [45]. The chemical adjustment of this molecular platform would allow for SMM arrangement in a three-dimensional molecular material [46] and for optimization of SMM behavior by means of the ligand change [47,48].

With this in mind, the present work is aimed at the synthesis and investigation of the molecular structure and magnetic properties of a new complex of Dy(III) with two ligand types, i.e., $[Dy(acac)_3(PyNO)]_2 \cdot 2CHCl_3$ (acac—acetylacetonate, PyNO—pyridine-N-oxide).

## 2. Materials and Methods

### 2.1. Synthesis

All chemicals were commercially purchased and were used without further purification. KOH (Sigma-Aldrich, St. Louis, MO, USA), $Dy(NO_3)_3 \cdot 5H_2O$ (Sigma-Aldrich), acetylacetone ($CH_3COCH_2COCH_3$) (Sigma-Aldrich), and pyridine-N-oxide ($C_5H_5NO$) (Sigma-Aldrich) were used in the work. For solution preparation, bidistilled water, absolute methanol, and trichloromethane were used. All the solvents were desiccated and additionally purified by distillation according to the procedures in [49].

Dysprosium acetylacetonate was obtained according to the procedure in [40].

Complex **1** was synthesized as follows: dysprosium acetylacetonate (0.12 mmol, 0.06 g) and pyridine-N-oxide (0.12 mmol, 0.0114 g) were dissolved in 40 mL of $CHCl_3$, then the mixture was filtered, stirred, covered by n-heptane layer, and left until white single crystals appeared. The yield was 13%.

### 2.2. IR Spectroscopy

The IR spectrum of **1** ($cm^{-1}$) was acquired using a Fourier-Transform Infrared spectrometer (Bruker Optik GmbH, Ettlingen, Germany) in the frequency range of 400–4000 $cm^{-1}$ in ATR mode at room temperature: 3117 (v.w.), 3073 (v.w.), 2984 (w.), 2922 (v.w.), 1596 (s.), 1546 (w.), 1513 (v.s.), 1469 (s.), 1416 (s.), 1358 (s.), 1259 (m.), 1219 (m.), 1191 (w.), 1179 (w.), 1150 (w.), 1073 (v.w.), 1014 (m.), 918 (m.), 836 (w.), 776 (m.), 763 (m.), 743 (v.s.), 673 (m.), 654 (s.), 554 (w.), 528 (m.), 499 (w.), 468 (w.).

### 2.3. Elemental Analysis

The elemental analysis was performed on a CHNS/O element analyzer vario MICRO cube (Elementar, Langenselbold, Germany).

For $C_{42}H_{54}Cl_6N_2O_{14}Dy_2$ (**1**) Found, %: C—37.38, H—4.01, Cl—15.75, N—2.09, O—16.60, Calculated, %: C—37.40, H—4.04, Cl—15.77, N—2.08, O—16.61.

*2.4. X-ray Analysis*

X-ray diffraction analysis of compound **1** was performed on a CCD diffractometer Agilent XCalibur with an EOS detector (Agilent Technologies UK Ltd., Yarnton, Oxfordshire, UK). Data collection, determination, and refinement of the unit cell parameters were performed with CrysAlis PRO software [50]. The single crystals were studied at 100 K. The structure was solved by the direct method. Full-matrix refinement of positions and thermal parameters of the non-hydrogen atoms was performed isotropically, followed by anisotropic refinement by the least-squares method (LSM). Positions of the hydrogen atoms were obtained from the difference synthesis and refined in the riding scheme.

Parameters of the unit cell and main crystallographic data are shown in Supplementary Materials Table S1, and some bond lengths and valence angles are presented in Supplementary Materials Table S2. The crystalline structure was deposited with the Cambridge Structural Database, CCDC 2237533, and it can be accessed at www.ccdc.cam.ac.uk/data_request/cif (accessed on 23 January 2023).

The powder XRD pattern for **1** was recorded at room temperature on an Aeris diffractometer (Malvern PANalytical B.V., Almelo, The Netherlands). The powder XRD measurement showed that polycrystalline sample **1** is a monophase crystalline material corresponding to the single crystal data (Figure S1).

*2.5. Magnetic Properties*

The DC and AC magnetic properties of complex **1** were analyzed using a vibrating sample magnetometer of a Cryogen Free Measurement System (CFMS, Cryogenic Ltd., London, UK). The temperature dependences of the magnetic moment M(T) were measured upon cooling at T = 2–300 K in a static magnetic field H = 5000 Oe. The cooling rate was 2 K/min. The field dependencies of the magnetic moment M(H) were obtained at temperatures 2, 4, 6, 8, and 10 K in the field range of 0–90,000 Oe. The field change rate was 5000 Oe/min.

The AC magnetic behavior of complex **1** was studied at T = 10–21 K in a 4 Oe oscillating field in the frequency range of 10–10,000 Hz in the absence and with the application of a DC magnetic field H = 1500 Oe.

All measurements were carried out on powder samples moistened with mineral oil Fomblin YR 1800 (Alfa Aesar, Heysham, United Kingdom) to prevent the orientation of some crystals in the DC magnetic field. The prepared samples were sealed in plastic bags. The magnetic susceptibility χ was determined taking into account the diamagnetic contribution of the substance, using the Pascal scheme, and considering the bag and mineral oil contribution.

*2.6. Computational Methodology*

All the multi-reference calculations were performed using the ab initio CASSCF/RASSI + SO/SINGLE(POLY)_ANISO approaches using the OpenMolcas program [51,52]. All calculations were based on the experimental X-ray structure of the binuclear complex **1**. The [ANO-RCC . . . 8s7p5d3f2g1h.] basis set for Dy atom, the [ANO-RCC . . . 3s2p1d] for N and O atoms, [ANO-RCC . . . 3s2p] for C atoms and [ANO-RCC . . . 2s] for H atoms have been employed. For calculations of single-ion properties of Dy(III) ions in complex **1**, we have replaced the second ion with the diamagnetic Lu(III) ion. The active space comprises CAS (9, 7) (where 9 represents the number of active f-electrons for Dy(III) ion in seven 4f-based active orbitals). The electronic configuration of the Dy(III) ion is $4f^9$, possessing a $^6H_{15/2}$ ground state. Using the active space comprised of CAS(9,7), we have computed 21 spin-free sextet states. All these computed spin-free states for the Dy(III) center were later mixed using the RASSI-SO module to compute the spin-orbit states. The SINGLE_ANISO module was used to calculate the magnetic anisotropy and

crystal field parameters. Resolution of identity Cholesky decomposition (RICD) was turned on to save disk space. Using the POLY_ANISO code, here we have simulated the dc magnetic properties (magnetic susceptibility and magnetization data) to extract the magnetic exchange interaction between the Dy(III) centers in complex **1**.

## 3. Results and Discussion

### 3.1. Synthesis

Acetylacetone (R=CH$_3$ in Scheme 1) is known [53,54] to exist in two isomeric forms: keto- and enol-tautomers (the latter in protonic solvents) and to easily coordinate in a chelate-like way the ion of a rare-earth metal through two donor oxygen atoms. This is accompanied by a shift in the band of C=O valence vibrations (a single band) of acac ($\nu$C=O 1650 cm$^{-1}$) (Supplementary Materials Figure S2) to lower frequencies ($\nu$(C=O) 1580–1583 cm$^{-1}$). In the IR spectra of these complexes, intense broad absorption bands are observed in the range of 3000–3600 cm$^{-1}$ as in [55], which is due to the presence of the crystallization water, and this is consistent with the X-ray data for [Dy(acac)$_3$(H$_2$O)$_2$]·2H$_2$O [40]. For this complex, in aprotic halogen-containing solvents, two PyNO molecules substitute for two coordinated water molecules (Scheme 2), which can be seen from the experimental IR spectrum of compound **1** (Supplementary Materials Figure S3).

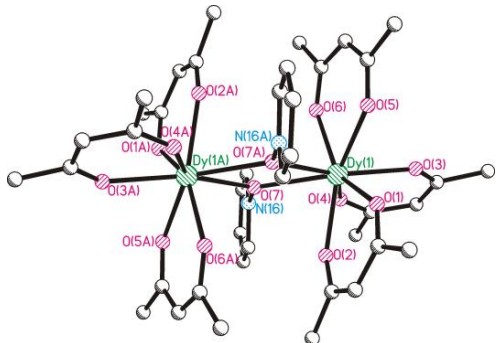

**Scheme 1.** Structures of isomeric forms of β-ketonates.

$$2[\text{Dy(acac)}_3(\text{H}_2\text{O})_2] + \underset{(\text{PyNO})}{\text{[PyNO]}} \xrightarrow{\text{CHCl}_3 / n\text{-heptane}} [\text{Dy(acac)}_3\text{PyNO}]_2 \cdot 2\text{CHCl}_3$$

**Scheme 2.** Synthesis of complex **1**.

### 3.2. X-ray Analysis

Complex **1** crystallizes in the monoclinic system, space group P2$_1$/c (Supplementary Materials Table S1). It is an electroneutral binuclear complex (Figure 1), with the oxygen atoms of two PyNO molecules being the bridges between two Dy ions, i.e., PyNO molecules link two Dy(III) ions in a μ$_2$ mode. Complex **1** is centrosymmetric; the asymmetric part of the unit cell involves one half of complex **1**, i.e., one Dy(III) atom, three acetylacetonate ligands, one PyNO molecule, and a chloroform solvate molecule. The intramolecular Dy . . . Dy distance is 4.264 Å.

**Figure 1.** Molecular structure of **1**. The hydrogen atoms are not shown for clarity, the carbon atoms are not marked. Color code: green—Dy; blue—nitrogen; red—oxygen; grey—carbon. Atoms labeled with the letter A are generated by the symmetry code (1—x, —y, 2—z).

Each Dy ion is surrounded by eight oxygen atoms (Figure 2). Each β–diketonate anion provides two donor O atoms with Dy–O distances ranging from 2.312(4) to 2.360(5) Å. The other two coordination sites of the Dy ion are occupied by two O atoms from the PyNO ligands with Dy–O distances of 2.425(5) and 2.514(5) Å. Selected bond lengths and angles are shown in Supplementary Materials Table S2.

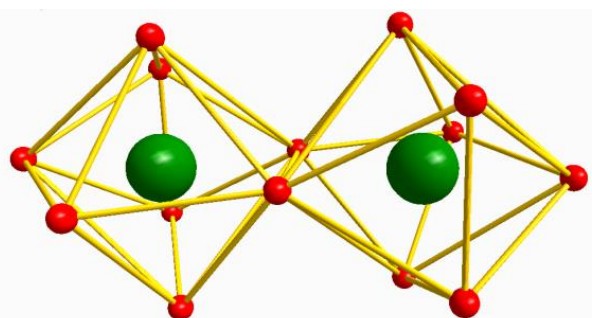

**Figure 2.** Local coordination environment of Dy ions in **1**. Color code: green—Dy; red—oxygen.

As follows from the SHAPE 2.1 program analysis [56], the Dy coordination polyhedron is a triangular dodecahedron ($D_{2d}$ symmetry) with CShM parameter 0.492 (Supplementary Materials Table S3), this being the essential distinction of complex **1** from two earlier studied binuclear complexes [Dy(hfac)$_3$(PyNO)]$_2$ having a similar structure [44]. In both hfac-based complexes, the local coordination environment of the Dy ions is a square antiprism.

Supplementary Materials Figure S4 shows the crystalline packing of **1**. The shortest interdimer distance Dy . . . Dy is 8.324 Å. The crystalline structure is stabilized by weak van der Waals bonds. All intermolecular contacts form through chloroform molecules. The contacts of the hydrogen atoms of acetylacetonate groups with the chlorine atom (C-H . . . Cl) and of the hydrogen atom of chloroform with the oxygen atoms of acetylacetonate groups (C-H . . . O) are shortened. Evidently, this is one reason for the crystals' decomposition upon a weak mechanical impact.

### 3.3. Magnetic Properties

### 3.3.1. Direct Current (dc) Magnetic Susceptibility of **1**

Temperature-dependent magnetic susceptibility measurements were performed in the range of 2–300 K under 0.5 T (Figure 3a). The room temperature $\chi_M T$ value of 27.97 cm$^3$ Kmol$^{-1}$ is found to be slightly lower than the calculated values for two non-interacting Dy(III) ions in the free-ion approximation ($^6$H$_{15/2}$, g$_J$ = 4/3 28.34). Upon decreasing the temperature, the $\chi_M$T values of **1** remain almost constant up to 50 K and decrease gradually up to 10 K, then fall abruptly and reach the minimum value of 14.63 cm$^3$ Kmol$^{-1}$ at 2 K. Comparison of the experimental and calculated dependences of the magnetic susceptibility at different values of $J_{ex}$ made it possible to estimate the value of the exchange coupling constant between Dy(III) ions in binuclear complex **1** (Figure 3a).

The magnetization (M) data of **1** were collected at 2, 4, 6, 8, and 10 K in the field ranges of 0–9 T (Figure 3b). The magnetization goes up consistently with the increased applied dc field, and it is saturated at 9 T. The magnetization values of **1** (10.36 Nβ) are lower than the theoretical saturation value (20 Nβ) expected for two isolated Dy(III) ions, which can be explained by the crystal-field splitting and magnetic anisotropy effects.

Magnetic hysteresis loops were measured at 2 K with magnetic field sweep rates of 0.3 T/min and 0.9 T/min (Supplementary Materials Figure S5). The coercive field depends on the sweep rate: 190 Oe at 0.3 T/min and 420 Oe at 0.9 T/min. This behavior is typical for single-molecule magnets.

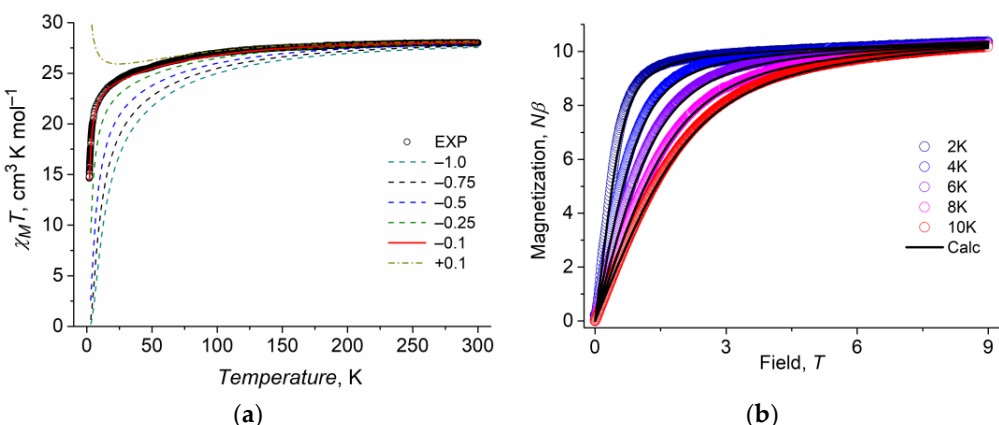

(**a**)  (**b**)

**Figure 3.** (**a**) DC magnetic susceptibilities for complex **1**. The solid red line represents the best POLY_ANISO simulated data within $J_{ex} = -0.1$ cm$^{-1}$, while the dashed lines represent the POLY_ANISO simulated data with different $J_{ex}$ values. (**b**) Field-dependent magnetization at 2, 4, 6, 8, and 10 K for complex **1**. The solid black lines represent the POLY_ANISO simulated data.

3.3.2. Dynamic (ac) Magnetic Properties of **1**

Frequency-dependent alternating-current (ac) magnetic susceptibility measurements for complex **1** were performed in the temperature range of 10–21 K under zero dc field (Figures 4a and S6a). A fit of the out-of-phase ac susceptibility dependence was performed with the Debye function. The position of the maximum at temperatures <10 K and >21 K is outside the CFMS vibrating magnetometer frequency range (10–10,000 Hz).

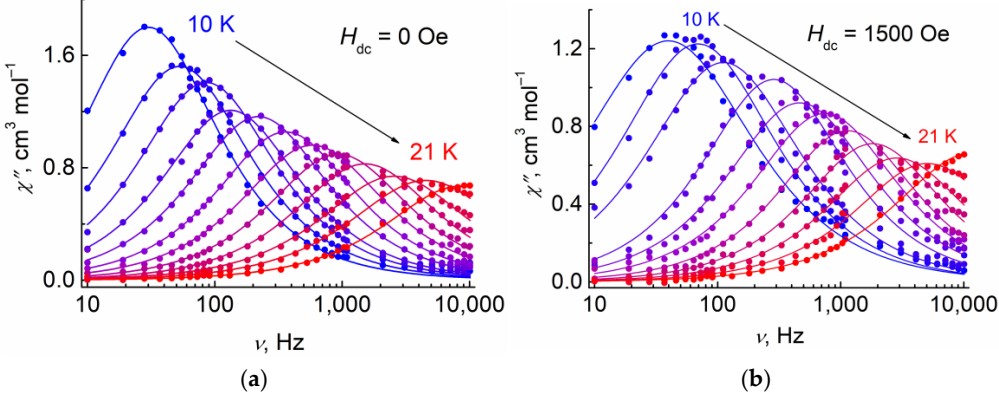

(**a**)  (**b**)

**Figure 4.** Frequency dependences of the out-of-phase ac susceptibility signals for complex **1** at zero (**a**) and 1500 Oe (**b**) dc fields. Dots are experimental data; lines are fits with the generalized Debye model.

The position of the maximum at temperatures of 10–21 K does not remain constant (Figure 4a), which indicates the absence of a quantum tunneling of magnetization (QTM) process at this temperature range. On the contrary, at these temperatures, the maximum shows a shift to higher frequencies with increasing temperature, with the maximum shifting beyond the CFMS magnetometer range (10–10,000 Hz) at T > 21 K.

At the applied dc field 1500 Oe, frequency dependences of the in-phase and out-of-phase ac susceptibility signals (Figures 4b and S6b) show similar relaxation behavior. The position of the maximum remains almost unchanged compared to at zero dc field (Figure S7).

The Arrhenius dependences of the natural log of the relaxation time $\tau$ vs. inverse temperature for relaxation processes at zero and 1500 Oe are shown in Figure 5. A fit of the experimental data was performed with Equation (1), the obtained best-fit parameters are listed in Table 1.

$$\tau^{-1} = \tau_0^{-1} \cdot \exp(-U_{\text{eff}}/k_B T) + CT^n \qquad (1)$$

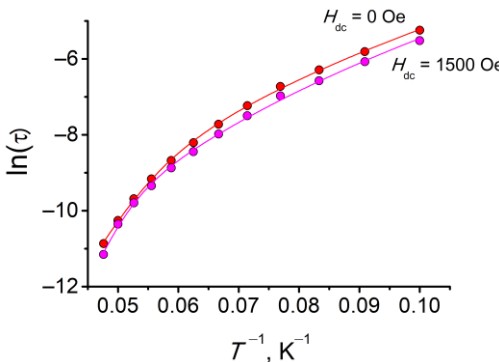

**Figure 5.** The $\ln(\tau)$ vs. $1/T$ dependencies for complex **1** at zero and 1500 Oe dc field. Dots are experimental data; lines are fits with Equation (1).

**Table 1.** Best-fit parameters with Equation (1) for the magnetic relaxation time for complex **1**.

| $H_{dc}$, Oe | 0 | 1500 |
|---|---|---|
| $\tau_0$, s | $3.15 \times 10^{-11}$ | $1.24 \times 10^{-12}$ |
| $U_{eff}$, K | 287 | 338 |
| $C_{Raman}$, $s^{-1} \cdot K^{-n}$ | $2.2 \times 10^{-4}$ | $1.5 \times 10^{-4}$ |
| $n_{Raman}$ | 5.93 | 6.20 |

Surprisingly, the relaxation process appeared to be field-independent, and its dependencies at zero and applied fields almost match (Figure 5, red and magenta lines). Both could be fitted with a combination of Orbach and Raman with close parameters (Table 1). It should be noted that a combination of direct and QTM does not provide adequate fits.

The intramolecular Dy-Dy distances in similar binuclear complexes are in a narrow range of 4.02–4.17 Å. The corresponding value in the complex under consideration goes beyond this interval and is equal to 4.26 Å, however, the DC magnetic properties are very similar in all the complexes. The $\chi_M T$ values at room temperature are similar for all the complexes. They are somewhat lower than the value calculated for the pair of non-reacting $Dy^{3+}$ ions (28.33 cm$^3$ K mol$^{-1}$) and are equal to 27.97 cm$^3$ K mol$^{-1}$ for our compound **1**, and equal to 26.38 and 27.23 cm$^3$ K mol$^{-1}$ for the compounds studied in [44]. This can be explained by the similar values of the exchange integrals determined in our work ($-0.1$ cm$^{-1}$) and in [44] ($-0.5$ and $-0.125$ cm$^{-1}$).

However, the AC magnetic properties differ dramatically. For example, the relaxation time $\tau$ at 10 K differs by orders of magnitude (5.2 ms in our work vs. 0.16 and 0.34 ms in Ref. [43]). The significant difference in the relaxation parameters is probably due to the different local coordination environments of the Dy ions. In our compound, the coordination polyhedron of the Dy ion is a triangular dodecahedron, whereas in the compounds studied in Ref. [44], the local coordination environment of the Dy ions is a square antiprism. Such differences in the AC magnetic properties could be also due to a rather short intermolecular distance between the Dy ions in the crystalline structure of the studied compound (8.32 Å), while for the formerly studied similar binuclear dimers, these distances lie in a higher range of 8.76–12.4 Å.

### 3.3.3. Theoretical Insight of the Dy Ions Magnetic Anisotropy in **1**

CASSCF/RASSI + SO/SINGLE_ANISO calculation of complex **1** was carried out to understand the nature of the magnetic anisotropy of the Dy ions and their magnetic behavior. The calculation was performed on the individual Dy center by replacing the other center with its diamagnetic analog Lu(III). Due to the centrosymmetric structure, we have observed the perfect coincidence of the computed energy diagrams and g-tensors values for Dy centers in **1**. The eight lowest Kramer's doublets (KDs) and *g*-tensors calculated for **1** using CASSCF/RASSI approach are summarized in Table 2.

**Table 2.** The ab initio calculated energy levels (cm$^{-1}$) and associated *g*-tensors of the eight lowest KDs for Dy ion in **1**.

| KD | Energy | $g_x$ | $g_y$ | $g_z$ |
|----|--------|-------|-------|-------|
| 1 | 0.0 | 0.003 | 0.007 | 19.666 |
| 2 | 162.0 | 0.340 | 0.407 | 16.020 |
| 3 | 236.9 | 3.871 | 4.262 | 13.147 |
| 4 | 275.8 | 6.851 | 5.849 | 3.310 |
| 5 | 308.9 | 3.271 | 5.709 | 12.247 |
| 6 | 372.6 | 0.032 | 0.080 | 18.336 |
| 7 | 433.8 | 0.006 | 0.055 | 16.535 |
| 8 | 493.7 | 0.023 | 0.066 | 18.388 |

The calculated effective $g_z$ component of the *g*-tensor is 19.666 for ground KD of Dy ion in **1** at $g_x$ and $g_y$ are almost equal to zero, which corresponds to the Ising type feature ($g_x = g_y \sim 0$; $g_z \sim 20$) for a pure $M_J = 15/2$ ground state; thus, Dy(III) centers have a significant axial anisotropy. Wave function decomposition analysis has shown that the ground KD composition is predominantly 94.5%$|\pm 15/2>$ with a small admixture from the 5.4%$|\pm 11/2>$ state (Table S4). The effective barrier 287 K (338 K at applied 1500 Oe field) (Table 1) of the thermally assisted Orbach relaxation mechanism obtained from the ln($\tau$) vs. 1/T dependencies are in satisfactory agreement with the energy gaps between the ground and second excitation states 341.1 K (236.9 cm$^{-1}$) (Table 2).

Magnetic relaxation pathways can be estimated on the basis of transition magnetic moments (Figure 6). For the most similar complexes [42–44], there is a correspondence between the U$_{\text{eff}}$ barrier value and the energy gap between the ground and the first excited state; it is also in agreement with the strong mixing of the wave function of the first excited KD and the high value of the QTM probability. For Dy ions in complex **1**, evidently, the relaxation proceeds through the second excited state, which correlates with the barrier values and calculated energy gaps, low TA-QTM probability for the first excited state, and relatively high purity of this state (80%$|\pm 13/2>$); moreover, it retains the strong axiality of the *g*-tensor that has an Ising type feature as the ground state (Table 2).

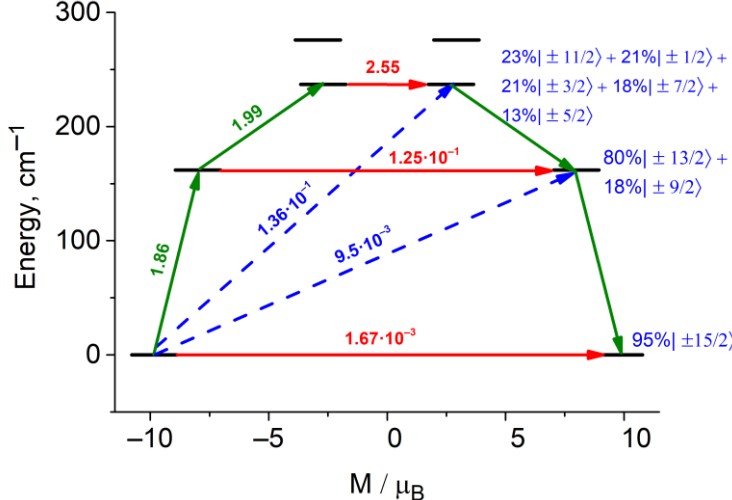

**Figure 6.** The computed possible magnetization relaxation pathways for one Dy ion in **1**. The red arrows show QTM and TA-QTM via ground and higher excited KD, respectively. The blue arrows show the Orbach process for relaxation. The green arrows show the mechanism of magnetic relaxation.

## 4. Conclusions

We report here the structure and magnetic characteristics of the first Dy(III) acetylacetonate dimer, with the Dy(III) atoms linked by two bridging pyridine-N-oxide molecules, which exhibits SMM properties. The presence of acetylacetonate ligands instead of hex-

afluoroacetylacetonates and other β-diketonates results in a change in the Dy(III) ion coordination environment toward a decrease in local symmetry. This does not substantially affect DC magnetic properties but manifests itself in dynamic characteristics (AC susceptibilities), namely, in an increase in the relaxation time.

The development of synthetic strategies based on the use of diketone fragments, which easily yield rare-earth SMMs, combined with ligands of another chemical origin is a promising trend for designing new rare-earth SMM structures and related materials with exciting characteristics by controlling parameters of the synthesis (pH, temperature, time, solvents, etc.).

**Supplementary Materials:** The following supporting information can be downloaded at https://www.mdpi.com/article/10.3390/magnetochemistry9040105/s1, Figure S1: Powder X-ray diffraction pattern of polycrystalline sample of 1; Figure S2: Experimental IR spectrum of acac; Figure S3: Experimental IR spectrum of complex 1; Figure S4: Crystalline packing of 1; Figure S5: Magnetic hysteresis loops at 2 K with magnetic field sweep rates of 0.3 T/min (blue line) and 0.9 T/min (red line); Figure S6: Frequency dependencies of the in-phase ac susceptibility for 1 at zero (a) and 1500 Oe (b) dc fields and temperatures from 10 to 21 K; Figure S7: Frequency dependencies of the out-of-phase ac susceptibility for 1 at T = 14 K and applied magnetic fields from zero to 4000 Oe; Table S1: Main crystallographic data for the crystal of complex 1; Table S2: Selected bond lengths and valence angles in complex 1; Table S3: The local symmetry of Dy(III) ions for 1 defined by the continuous shape measure (CShM) analysis with SHAPE 2.1 software; Table S4: SINGLE_ANISO computed wave function decomposition analysis for the lowest KDs of Dy(III) ion in 1. Only main contributions (>10%) are shown.

**Author Contributions:** V.P.S.: investigation (synthesis); G.V.S.: investigation (X-ray diffraction analysis); D.V.K. and E.A.Y.: investigations (quantum chemical calculations); A.I.D., M.V.Z. and R.B.M.: investigations (magnetic properties); N.A.S.: conceptualization, methodology, writing the manuscript (original draft preparation); S.M.A.: writing the manuscript (review and editing). All authors have read and agreed to the published version of the manuscript.

**Funding:** The work has been performed with the financial support of the Ministry of Science and Higher Education of the Russian Federation, Project "Fundamentals of spin technologies and directed design of "smart" polyfunctional materials for spintronics and molecular electronics" (Agreement No. 075-15-2020-779).

**Institutional Review Board Statement:** Not applicable.

**Informed Consent Statement:** Not applicable.

**Data Availability Statement:** Not applicable.

**Acknowledgments:** The authors appreciate the help of M.Z. Aldoshina. Elemental analysis was performed using the equipment of the Multi-user Analytical Center of FRC PCP MC RAS. The APC was funded by MDPI-Magnetochemistry.

**Conflicts of Interest:** The authors declare no conflict of interest.

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
