# Peer review of "Zero-Field Slow Magnetic Relaxation in Binuclear Dy Acetylacetonate Complex with Pyridine-N-Oxide"

_magnetochemistry, doi:10.3390/magnetochemistry9040105_

Round 1

Reviewer 1 Report (Previous Reviewer 2)

the authors have revised the manuscript carefully, it could be accepted now.

Author Response

Dear Reviewer, your valuable comments and suggestions have improved our article. We would like to thank you for your time. 

Reviewer 2 Report (New Reviewer)

In this manuscript the authors reported a pyridine-N-oxide bridged binuclear Dy(III) complexes with acetylacetonates as co-ligands. This complex exhibits interesting magnetic relaxation behavior.
The publication of this manuscript in Magnetochemistry is recommended. However, the revisions should be made before publication.
1.       For theoretical calculations, the calculated energy levels and corresponding g values should be given, then discuss them. For the fitting of dc data, Jdip? 2.       The magnetic hysteresis loop should be measured. 3.       For keywords,”Single molecular magnets” and “SMM” are the same meaning. 4.       “rare-earth” and “ lanthanide” should be unified. 5.       page 1, line 13, “like in the case of La [13] “?

Author Response

Reviewer 2  
Comments and Suggestions for Authors

In this manuscript the authors reported a pyridine-N-oxide bridged binuclear Dy(III) complexes with acetylacetonates as co-ligands. This complex exhibits interesting magnetic relaxation behavior.

The publication of this manuscript in Magnetochemistry is recommended. However, the revisions should be made before publication. 1.       For theoretical calculations, the calculated energy levels and corresponding g values should be given, then discuss them. For the fitting of dc data, Jdip? 2.       The magnetic hysteresis loop should be measured. 3.       For keywords,”Single molecular magnets” and “SMM” are the same meaning. 4.     “rare-earth” and “ lanthanide” should be unified. 5.       page 1, line 13, “like in the case of La [13] “?

our response:

The authors agree with the Reviewer’s comment, and the corresponding changes have been made in the new version of the manuscript, namely:

  1. According to our estimates in the POLY_ANISO program, the dipolar interactions between the spin centers do not exceed 0.00167 cm^-1, which are quite small values even taking into account the small value of the exchange interaction (-0.1 cm^-1 ); in this regard, we neglected them while describing the magnetic properties.
  2. The magnetic hysteresis loops were measured. The obtained data are presented in SI (see Fig SI4 in Supplementary Information)

Items 3-5.  The corresponding changes have been made in the new version of the manuscript.

Reviewer 3 Report (New Reviewer)

One new complex [Dy(C5H7O2)3(C5H5NO)]2.2CHCl3 was synthesized and studied by IR, elemental analyses and single crystal analyses. Additionally magnetic properties of the compound were measured.

The manuscript was written well and it could be published with minor revision. I can only recommend authors to give the scheme how the complex 1 was synthesized.

Author Response

Reviewer 3 
Comments and Suggestions for Authors

One new complex [Dy(C5H7O2)3(C5H5NO)]2.2CHCl3 was synthesized and studied by IR, elemental analyses and single crystal analyses. Additionally magnetic properties of the compound were measured.

The manuscript was written well and it could be published with minor revision. I can only recommend authors to give the scheme how the complex 1 was synthesized.

our response:

The corresponding scheme is presented in the new version of the manuscript.

Round 2

Reviewer 2 Report (New Reviewer)

In the revised manuscript, the authors almost addressed my concerns, the paper could be accepted now.

This manuscript is a resubmission of an earlier submission. The following is a list of the peer review reports and author responses from that submission.

Round 1

Reviewer 1 Report

This contribution reports on a dinulear Dy(III) complex with pyridine-N-oxide as bridging ligand and acac ligands as anions. Preparation, crystal structure, magnetic studies are described, some theoretical calculations are also mentioned. The results are discussed in relation with the data reported for an homologous complex with hfac instead of acac anions.

This work is rather superficial and routine; no really new or important information is given. It is unfortunate that some observations are not questioned and explored further (see comments below). Today, the observation of just a slow relaxation phenomenon can no longer justify publication. And the characteristics identified in the present work are far from deserving of attention. For this reason, I do not recommend publication of this work in Magnetochemistry.

The weak points:

This paper reports on a dinuclear Dy(III) complex with pyridine oxide as bridging lingand, however, the experimental section mentions the use of mercaptopyridine-N-oxide as the reagent. Is this correct ? If so, the transformation must be underlined and explained.

Moreover, macroscopic characterization of the compound must be provided, this includes chemical analysis (C,H,N, and S if mercapto derivative was involved) and powder X-ray diffraction data.

The magnetic behaviors in fig 4 are depicted with calculated behavior. Why is this not commented? and what is the reference Hamiltonian?

From the caption we learn that the exchange interaction between the Dy centers is about -0.1 cm-1, which is a reasonable value but supposes that the crystal field for the Dy is properly reproduced. Therefore, the "best fit" includes two distinct contributions, that of the crystal field and that of the exchange interaction, with no way to evaluate the accuracy of each of these contributions. The authors could have studied the homologous compound of Gd to validate the magnitude of the exchange interaction between Ln.

The AC behavior clearly show two relaxations feature but what is the origin of them? Often this is found when 2 Ln centers have different relaxations characteristics due to their distinct coordination speher; but this can’t apply here because the Ln centers are identical based on the crystal structure.

It could have been interesting to know the behavior of a single Dy center in such a compound; this could be obtained in a complex formed with a mixture of Y(III) and about 10% of Dy(III). This would discard any dipolar contribution die to the proximity of the two Dy(III) in the reported complex.

Reviewer 2 Report

1.     There are many typesetting errors, such as “IRspectrumof1(cm-1) was…” The authors should check the full work.

2.     “In IR spectra of these 155 complexes, intense broad absorption bands are observed in the range of 3000 cm−1 - 3600 156 cm−1, which is due to the presence of the crystallization water,” This part should be updated the refs, such as Inorganics, 10(2022) 202 and Micropor. Mesopor. Mat, 341(2022) 112098. “In particular, in mononuclear SMM, QTM contribution in total relaxation increases with the temperature decrease.” This also be added some updated refs, such as J. Solid State Chem. 318(2023) 123713

3. In Fig. 1, please add the symmetric codes for the related atoms.

4. The PXRD and IR data should be checked for the purity of the samples.

5. The authors have stated “The magnetization (M) data of 1 were collected at 2, 4 and 6 K in the field ranges of 0–5T (Fig.4b).” Why the authors only not collect at 10K, and 10T?

Reviewer 3 Report

This manuscript by Shtefanets et al shows the efforts of their synthetic and SMM studies on dinuclear Dy complex bridged using N-pyridine oxide and surprisingly the complex showed dual relaxations. Even though this work is interesting to see, the authors need to clarify furthermore flaws in their work, and further more resaearch to be considered for this manuscript. In addition, there are very small mistakes in the draft and research, which need to be checked again.

*Importantly, authors need to proofread the whole manuscript, as the whole manuscript needs a lot of spaces between the words.
*Spelling mistake in the first paragraph - line 31 should have polyoxometallates and phthalocyanines
*Line 41-43. The concept of increasing nuclearity was an old myth and was with transition metal ions. Not in Lanthanide SMMs.
*Line 67 - "forming coatings on the adapted supports" is not clear
*Line 87 - mercaptopyridine-N-oxide or pyridine-N-oxide ?
*Any confirmation that the sample is pure? PXRD would be a perfect way to check if the whole compound obtained is the same as crystalline. Authors should provide the PXRD and Elemental analysis to confirm the purity. 
*Line 165- binuclear dimer and just dimer mean the same
*Why Just 10 K, it shows probably 12 or 14 K will show some more peaks for LF. 
*How did the authors decide 1300 was the Optimal field? using HF?
*It looks like the maxima at 10 K shifted right for LF upon applying Optimal Field. Why did this happen?
*what is the reason behind the two relaxations? Further more analysis is needed. Probably there might be an impurity. Should be confirmed with PXRD.